# Study on Paramagnetic Interactions of (CH_3_NH_3_)_2_CoBr_4_ Hybrid Perovskites Based on Nuclear Magnetic Resonance (NMR) Relaxation Time

**DOI:** 10.3390/molecules24162895

**Published:** 2019-08-09

**Authors:** Ae Ran Lim, Sun Ha Kim

**Affiliations:** 1Analytical Laboratory of Advanced Ferroelectric Crystals, Jeonju University, Jeonju 55069, Korea; 2Department of Science Education, Jeonju University, Jeonju 55069, Korea; 3Korea Basic Science Institute, Seoul Western Center, Seoul 120-140, Korea; 4Department of Chemistry, Kyungpook National University, Daegu 41566, Korea

**Keywords:** organic/inorganic hybrid materials, structure, dynamics, (CH_3_NH_3_)_2_CoBr_4_, MAS/NMR

## Abstract

The thermal properties of organic–inorganic (CH_3_NH_3_)_2_CoBr_4_ crystals were investigated using differential scanning calorimetry and thermogravimetric analysis. The phase transition and partial decomposition temperatures were observed at 460 K and 572 K. Nuclear magnetic resonance (NMR) chemical shifts depend on the local field at the site of the resonating nucleus. In addition, temperature-dependent spin–lattice relaxation times (T_1ρ_) were measured using ^1^H and ^13^C magic angle spinning NMR to elucidate the paramagnetic interactions of the (CH_3_NH_3_)^+^ cations. The shortening of ^1^H and ^13^C T_1ρ_ of the (CH_3_NH_3_)_2_CoBr_4_ crystals are due to the paramagnetic Co^2+^ effect. Moreover, the physical properties of (CH_3_NH_3_)_2_CoBr_4_ with paramagnetic ions and those of (CH_3_NH_3_)_2_CdBr_4_ without paramagnetic ions are reported and compared.

## 1. Introduction

Hybrid organic–inorganic compounds based on perovskite structures are currently attracting an increased amount of interest owing to their potential as substitutes for perovskite solar cells [1,2,3,4,5,6,7,8,9,10]. However, the toxicity and chemical instability of perovskites continue to be the major problems associated with their use in solar cells. Compounds in the (CH_3_NH_3_)_2_*MX*_4_ family (where *M* is the transition metal and *X* is halide) exhibit a variety of physical properties [1,11]. Ions of the transition metal *M* are located in the tetrahedral structure formed by the halogen ions *X*, and lie in the planes bridged by the (CH_3_NH_3_)^+^ cations [12]. These crystals have a layered structure and exhibit quasi-, two-dimensional magnetic properties. Most recently, electrochemical oxygen evolution of (CH_3_NH_3_)_2_CoBr_4_, a lead-free cobalt-based perovskite, has been reported by Babu et al. [13]. The (CH_3_NH_3_)_2_CoBr_4_ crystal belongs to the (CH_3_NH_3_)_2_*MX*_4_ series and the family of hybrid organic–inorganic compounds in which (CH_3_NH_3_)^+^ cations are connected via a bridge structure between the planes that contain the Co^2+^ ions. At room temperature, the (CH_3_NH_3_)_2_CoBr_4_ crystal structure has monoclinic symmetry and belongs to the space group *P2_1_/c*, with lattice constants a = 7.9782 Å, b = 13.1673 Å, c = 11.2602 Å, and ß = 96.3260° [14]. The unit cell contains four formula units and four magnetic Co^2+^ ions. The (CoBr_4_)^2−^ units are surrounded by seven (CH_3_NH_3_)^+^ cations, and two different crystallographic (CH_3_NH_3_)^+^ cations exist. Although the tetrahedral anion exhibits only C_1_ symmetry, the deviation from an idealized tetrahedral symmetry is small. The NH_3_^+^ polar heads of the chains connect the isolated (CoBr_4_)^2−^ tetrahedral structure with weak N‒H···Br hydrogen bonds. On the other hand, (CH_3_NH_3_)_2_CdBr_4_ crystals at room temperature have a monoclinic structure and belong to the space group *P2_1_/c* with lattice constants a = 8.1257 Å, b = 13.4317 Å, c = 11.4182 Å, ß = 96.1840°, and Z = 4 [15,16]. The structure of this crystal is very similar to that of the (CH_3_NH_3_)_2_CoBr_4_. Until now, the phase transition temperature, thermal property, and paramagnetic interactions of (CH_3_NH_3_)_2_CoBr_4_ have not been studied in full. The paramagnetic ions of the lead-free perovskite are eco-friendly, which is important for application to solar cells.

The present study was conducted to investigate the thermodynamic properties of the (CH_3_NH_3_)_2_CoBr_4_ crystal using differential scanning calorimetry (DSC), thermogravimetric analysis (TGA), and optical polarizing microscopy. Additionally, the nuclear magnetic resonance (NMR) chemical shifts and spin–lattice relaxation times T_1ρ_ in the rotating frame of (CH_3_NH_3_)_2_CoBr_4_ were obtained using ^1^H magic angle spinning (MAS) NMR and ^13^C cross-polarization (CP)/MAS NMR methods at several temperatures to probe the local environments and study the roles of the (CH_3_NH_3_)^+^ cations. Moreover, the physical properties of (CH_3_NH_3_)_2_CoBr_4_ including paramagnetic ions and (CH_3_NH_3_)_2_CdBr_4_ excluding paramagnetic ions were obtained from previous reports [17], and used as a comparison to understand the effects of Co^2+^ and Cd^2+^ ions.

## 2. Results and Discussion

TGA and DSC measurements were obtained to understand the thermal stability, structural phase transitions, and melting temperatures. The TGA and DSC curves of (CH_3_NH_3_)_2_CoBr_4_ are plotted within the temperature range of 300–770 K, as shown in Figure 1 and Figure 2. The transformation anomaly at 460 K (=T_C_) in the DSC curve is related to the phase transition. The mass loss of 3.89% occurs at approximately 572 K (=T_d_), and is ascribed to the onset of partial thermal decomposition. The compound (CH_3_NH_3_)_2_CoBr_4_ loses its crystallization at increased temperatures. When comparing the experimental TGA results and possible chemical reactions, the solid residue is calculated on the basis of Equations (1) and (2):
(CH_3_NH_3_)_2_CoBr_4_→(CH_3_NH_2_·HBr)_2_CoBr_2_→(CH_3_NH_2_)_2_CoBr_2_ (s) + 2HBr (g)(1)
Residue:[(CH_3_NH_2_)_2_CoBr_2_ (M = 280.857 g)]/[(CH_3_NH_3_)_2_CoBr_4_ (M = 442.681 g)] = 63.4% (CH_3_NH_3_)_2_CoBr_4_→(CH_3_NH_2_·HBr)_2_CoBr_2_→CoBr_2_ (s) + 2(CH_3_NH_2_·HBr) (g)(2)Residue:[CoBr_2_ (M = 218.741 g)]/[(CH_3_NH_3_)_2_CoBr_4_ (M = 442.681 g)] = 49.4 %

The mass loss of 37% near 669 K is likely attributable to the decomposition of the 2HBr moieties. Moreover, the mass loss near 700 K reaches 48.81%. These results are consistent with the TGA results reported by Babu et al. [13]. By the end, only CoBr_2_ remains. The solid-state decomposition is essentially one of the chemical reactions that occur at the surface. The second stage is associated with the thermal decomposition of (CH_3_NH_3_)_2_CoBr_4_ to CoBr_2_. Optical polarizing microscopy showed that the crystals have a seagrass color at room temperature. The color of the crystal does not vary as the temperature increases, and the crystal starts to melt at temperatures above T_d_, as indicated at the surface. From the TGA and DSC results, the phase transition temperature is 460 K, and the partial decomposition temperature is at 572 K. The high-temperature phenomenon above T_d_ is not related to a physical change, such as structural phase transitions, but is instead related to chemical changes, such as thermal decomposition.

The temperature-dependent ^1^H-NMR spectrum of (CH_3_NH_3_)_2_CoBr_4_ is obtained to understand and analyze its structure. All recorded spectra contain only one resonance line, and Figure 3 shows the spectrum at 410 K. The spinning sideband for ^1^H in CH_3_ is marked with open circles, and that for ^1^H in NH_3_ is marked with asterisks. The ^1^H resonance line has an asymmetric shape, and the full-width at half maximum (FWHM) values on the left and right sides are not equal. The asymmetric line shape is attributed to the overlapping lines of the two ^1^H in the (CH_3_NH_3_)^+^ cations. The ^1^H-NMR chemical shift of δ = −0.3 ppm is due to the CH_3_, while the ^1^H-NMR chemical shift of δ = 4.2 ppm is due to the NH_3_. The ^1^H-NMR chemical shifts for the two ^1^H in the (CH_3_NH_3_)^+^ cations are temperature-independent. They remain quasi-constant with increasing temperature, indicating that the structural environment of ^1^H in the CH_3_ and NH_3_ groups does not change.

Figure 4 shows the recovery traces for the ^1^H resonance lines for delay times that range from 1 μs to 20 ms at 300 K. Herein, the arrows mark the resonance lines at each delay time, while the other resonance lines are the sidebands. The T_1ρ_ values are obtained from the intensities of the magnetization recovery curves with respect to the delay time. The recovery traces are described by a simple mono-exponential function [18,19,20].
P(*τ*) = P(0) exp(−*τ* /T_1ρ_)(3)
where P(*τ*) is the NMR signal intensity measured after recovery time *τ*, and P(0) is the total nuclear magnetization of the protons at thermal equilibrium. This analysis method is used to obtain the T_1ρ_ values for the proton in the (CH_3_NH_3_)^+^ cations. However, the ^1^H T_1ρ_ values for CH_3_ and NH_3_ are indistinguishable owing to the overlapping responses of the two protons. The ^1^H T_1ρ_ values for (CH_3_NH_3_)_2_CoBr_4_ obtained herein and the corresponding values for (CH_3_NH_3_)_2_CdBr_4_ reported previously [17] are shown in Figure 5 as a function of the inverse temperature. In the case of (CH_3_NH_3_)_2_CoBr_4_, the ^1^H T_1ρ_ values increased rapidly near 210 K, and those at high temperatures are almost continuous; the T_1ρ_ value at 180 K is 76 μs and that at 300 K is 10 times longer than that at 180 K. The T_1ρ_ value is very short at low temperatures, and thus indicates rapid energy transfer from the nuclear spin system to the surrounding environment. On the other hand, the ^1^H T_1ρ_ values are obtained for each proton in CH_3_ and NH_3_ in the case of (CH_3_NH_3_)_2_CdBr_4_ as a function of reciprocal temperature. Herein, the T_1ρ_ values for the two protons of the (CH_3_NH_3_)^+^ cations are nearly the same within experimental error. The T_1ρ_ values of ^1^H in the CH_3_ and NH_3_ ions abruptly decrease at approximately 360 K. The ^1^H T_1ρ_ value of (CH_3_NH_3_)_2_CoBr_4_ including the paramagnetic ions is very short, whereas that of (CH_3_NH_3_)_2_CdBr_4_ excluding paramagnetic ions is very long.

The local environment of the carbons in (CH_3_NH_3_)_2_CoBr_4_ was studied by ^13^C MAS NMR, and the corresponding ^13^C-NMR chemical shifts are shown in Figure 6. Attention was paid to ^13^C-NMR, which should be a sensitive probe of the local environment and of the cation dynamics.

The ^13^C-NMR spectrum at 300 K in (CH_3_NH_3_)_2_CoBr_4_ shows two signals at the chemical shifts of δ = 68.3 ppm and δ = 117.9 ppm with respect to TMS [21]. The ^13^C-NMR spectrum consists of two lines that correspond to a-CH_3_ and b-CH_3_. The signals respectively represent the methyl carbons in the two crystallographically different a-CH_3_ and b-CH_3_. The ^13^C-NMR chemical shifts of the two compounds of (CH_3_NH_3_)_2_CoBr_4_ and (CH_3_NH_3_)_2_CdBr_4_ are shown in Figure 7 as a function of temperature. The ^13^C-NMR chemical shifts vary significantly with temperature. Specifically, the ^13^C-NMR chemical shifts in the case of (CH_3_NH_3_)_2_CoBr_4_ decrease slowly and monotonically as a function of temperature. Conversely, the ^13^C-NMR spectrum at 300 K in (CH_3_NH_3_)_2_CdBr_4_ shows two signals at chemical shifts of δ = 27.9 ppm and δ = 29.3 ppm. The ^13^C-NMR chemical shifts of the crystallographically different a-CH_3_ and b-CH_3_ slowly and monotonously increase as a function of temperature. The ^13^C chemical shifts of the CH_3_ groups differ between the two compounds. Generally, the paramagnetic contribution to the NMR shift is responsible for the NMR spectra. The ^13^C chemical shift of (CH_3_NH_3_)_2_CoBr_4_, which contains paramagnetic ions, was significantly different to that of (CH_3_NH_3_)_2_CdBr_4_, which does not contain paramagnetic ions. The differences in the ^13^C chemical shifts could potentially be due to differences in the electron structures of the metal ions.

To determine the ^13^C T_1ρ_, nuclear magnetization was measured as a function of the delay time. The signal intensities of the nuclear magnetization recovery curves are fitted by the mono-exponential function of Equation (3). From these results, T_1ρ_ values were obtained for the carbons in (CH_3_NH_3_)_2_CoBr_4_ and (CH_3_NH_3_)_2_CdBr_4_ as a function of the inverse temperature, as shown in Figure 8. In the case of (CH_3_NH_3_)_2_CoBr_4_, the T_1ρ_ values of ^13^C show a minimum value near 330 K, while the T_1ρ_ value abruptly decreases above 410 K. The T_1ρ_ values for a-CH_3_ and b-CH_3_ are also very similar and of the order of 10 ms. Conversely, the variation of T_1ρ_ with temperature in the case of (CH_3_NH_3_)_2_CdBr_4_ exhibits a minimum near 250 K for a-CH_3_ and b-CH_3_, respectively, and T_1ρ_ decreases abruptly above 360 K. The presence of these minima are attributed to the effects of the reorientation of (CH_3_NH_3_)^+^ cations. From the ^13^C T_1ρ_ curves, the relaxation processes of (CH_3_NH_3_)_2_CdBr_4_ are affected by molecular motion described by the Bloembergen–Purcell–Pound (BPP) theory [22]. The experimental values of T_1ρ_ are explained by the correlation time τ_C_ for molecular motion based on the BPP theory [22,23],
(1/T_1ρ_) = 0.05(*μ*_o_/4π)^2^[γ_H_^2^ γ_C_^2^ ħ^2^/*r*^6^][4F*_a_* + F*_b_* + 3F*_c_* + 6F*_d_* + 6F*_e_*](4) where
F*_a_* = τ_C_/[1 + ω_1_^2^τ_C_^2^]F*_b_* = τ_C_/[1 + (ω_H_ ‒ ω_C_)^2^τ_C_^2^]F*_c_* = τ_C_/[1 + ω_C_^2^τ_C_^2^]F*_d_* = τ_C_/[1 + (ω_H_ + ω_C_)^2^τ_C_^2^]F*_e_* = τ_C_/[1 + ω_H_^2^τ_C_^2^].
where *μ*_o_ is the permeability, γ_H_ and γ_C_ are the respective gyromagnetic ratios for the ^1^H and ^13^C nuclei, *r* is the distance of H–C, ħ = h/2π, and ω_H_ and ω_C_ are the respective Larmor frequencies of ^1^H and ^13^C.

On the other hand, the relaxation processes of (CH_3_NH_3_)_2_CoBr_4_ with the paramagnetic Co^2+^ ions are affected by the molecular motion described by the Solomon equation [24]. When paramagnetic ions exist, the T_1ρ_ are represented by τ_C_, as presented in [24]
(1/T_1ρ_) = (1/15)(*μ*_o_/4π)^2^[γ_I_^2^γ_e_^2^*μ*_B_^2^S(S+1)/*r*^6^][4G*_a_* + G*_b_* + 3G*_c_* + 6G*_d_* + 6G*_e_*](5) where
G*_a_* = τ_C_/[1 + ω_1_^2^τ_C_^2^]G*_b_* = τ_C_/[1 + (ω_C_ ‒ ω_e_)^2^τ_C_^2^]G*_c_* = τ_C_/[1 + ω_C_^2^τ_C_^2^]G*_d_* = τ_C_/[1 + (ω_C_ + ω_e_)^2^τ_C_^2^]G*_e_* = τ_C_/[1 + ω_e_^2^τ_C_^2^].

Here, γ_e_ is the gyromagnetic ratio of the electron, S is the total spin quantum number of the paramagnetic ion, and ω_e_ is the Larmor frequency of the electron. Additionally, ω_1_ is the angular frequency at the spin-lock field; 59.52 kHz for (CH_3_NH_3_)_2_CoBr_4_ and 67.56 kHz for (CH_3_NH_3_)_2_CdBr_4_. The T_1ρ_ exhibits a minimum when ω_1_τ_C_ = 1. Based on this condition, the coefficients of Equations (4) and (5) which are dependent on ω_1_, ω_H_, and ω_C_, can be obtained. Furthermore, the value of τ_C_ can be calculated, and its temperature dependence follows a simple Arrhenius expression [22] according to,
τ_C_ = τ_o_exp(−E_a_/RT)(6)
where τ_o_ is the preexponential factor, T is the temperature, R is the gas constant, and E_a_ is the activation energy. The activation energies for the tumbling motion of a-CH_3_ and b-CH_3_ in the case of (CH_3_NH_3_)_2_CoBr_4_ are obtained from the log τ_C_ vs. 1000/T curve, and are respectively equal to 24.51 ± 0.99 kJ/mol and 23.25 ± 1.30 kJ/mol, whereas the corresponding values in the case of (CH_3_NH_3_)_2_CdBr_4_ are 8.18 ± 0.37 kJ/mol and 7.65 ± 0.21 kJ/mol (see Figure 9). When paramagnetic Co^2+^ ions exist, 1/τ_C_ = 1/τ_r_ + 1/τ_M_ + 1/τ_e_, where τ_r_, τ_M_, and τ_e_, are the rotational correlation time, exchange correlation time, and electronic relaxation correlation time, respectively. The τ_r_ can represent molecular motion. For (CH_3_NH_3_)_2_CdBr_4_, there is no chemical exchange or paramagnetic terms, and so τ_C_ can directly reflect the molecular motion. In the case of (CH_3_NH_3_)_2_CoBr_4_, τ_e_ dominates the total correlation time, and thus, τ_C_ is not directly related to molecular motion.

## 3. Materials and Methods

The (CH_3_NH_3_)_2_CoBr_4_ single crystals were grown based on the slow evaporation of an aqueous solution with a 2:1 ratio of CH_3_NH_2_∙HBr and CoBr_2_ at 300 K. Single crystals have a diamond shape and seagrass color.

The thermal properties and phase transition temperature were measured using a TGA (TA, DSC 25) instrument at a heating rate of 10 °C/min. The TGA and DSC curves were measured in an N_2_ atmosphere, and the mass of the powder sample used in the experiment was 9.22 mg.

The solid-state MAS NMR spectra and the spin–lattice relaxation time T_1ρ_ in the rotating frame of (CH_3_NH_3_)_2_CoBr_4_ crystals were recorded on a Bruker 400 DSX NMR spectrometer (Bruker, Leipzig, Germany) at the Korean Basic Science Institute at the Western Seoul Center. Solid samples were inserted into 4 mm diameter zirconia rotors. The samples were spun at a sufficient speed to avoid spinning sidebands overlapping. The chemical shifts were defined with respect to tetramethylsilane (TMS). The ^1^H T_1ρ_ values were measured using a π/2-*τ* pulse sequence by varying the duration of the spin-locking pulses. The ^13^C T_1ρ_ values were measured based on the variation of the duration of the ^13^C spin-locking pulse. The usual experimental approach assumes the use of cross-polarization from protons to enhance the ^13^C sensitivity. The widths of the π/2 pulses for ^1^H and ^13^C were 4.1 μs and 4.2 μs, respectively. The T_1ρ_ values were measured in the temperature range of 180–430 K due to limitations of the experimental equipment associated with the measurements of the spectra and T_1ρ_ outside of this range. The sample temperatures were held constant by controlling the helium gas flow and the heater current [25,26].

## 4. Conclusions

The thermal properties and phase transition temperature of (CH_3_NH_3_)_2_CoBr_4_ crystals grown based on the slow evaporation method were investigated with TGA, DSC, and optical polarizing microscopy. The phase transition and partial decomposition temperatures were observed at 460 K and 572 K, respectively. The high-temperature phenomenon above 572 K was not related to a physical change like the structural phase transition. Instead, it was related to a chemical change, such as thermal decomposition.

The paramagnetic interactions of (CH_3_NH_3_)_2_CoBr_4_, associated with the role of the (CH_3_NH_3_)^+^ cations were studied by ^1^H-NMR and ^13^C-NMR as a function of temperature. The ^1^H and ^13^C MAS NMR were used to probe the dynamics of cations in (CH_3_NH_3_)_2_CoBr_4_ and (CH_3_NH_3_)_2_CdBr_4_. The chemical shift by the MAS NMR depended on the local field at the site of the resonating nucleus in crystals. The effect of these crystals on the ^1^H and ^13^C-NMR chemical shifts was investigated using temperature-dependent NMR experiments. The contributions to the ^13^C-NMR chemical shifts are correlated with the distribution of spin density in the ligand moiety.

The temperature dependence of the T_1ρ_ values for ^1^H reflect the modulation of the inter-NH_3_ and inter-CH_3_ dipolar interactions due to the (CH_3_NH_3_)^+^ cations. The variation of T_1ρ_ for ^13^C yielded a minimum, and it is apparent that the T_1ρ_ values for ^13^C are governed by tumbling motions. Moreover, the paramagnetic dopant led to the shortening of their T_1ρ_ values. Accordingly, it has been shown that the T_1ρ_ value is inversely proportional to the square of the magnetic moment of the paramagnetic ion [27]. The T_1ρ_ values of ^1^H and ^13^C of the (CH_3_NH_3_)_2_CoBr_4_ crystals, which contain paramagnetic ions, are much shorter than those of the (CH_3_NH_3_)_2_CdBr_4_ crystals, which do not contain paramagnetic ions.

The (CH_3_NH_3_)_2_CoBr_4_ and (CH_3_NH_3_)_2_CdBr_4_ crystals are of the (CH_3_NH_3_)_2_*MX*_4_ type, whereas their individual dynamics differ significantly from the dynamic of the cations. The differences between the T_1ρ_ of the (CH_3_NH_3_)_2_*M*Br_4_ crystals (*M* = Co and Cd) are due to the differences between the electron structures of their Co^2+^ and Cd^2+^ ions. These ions screen the nuclear charge from the motion of the outer electrons. The Co^2+^ has unpaired *d* electrons, whereas Cd^2+^ has filled *d* shells. Their NMR properties stem from the differences between the chemical properties of paramagnetic Co^2+^ and non-paramagnetic Cd^2+^ ions. Furthermore, the NMR relaxation of diamagnetic Cd^2+^ ions is most probably driven by dipolar interactions, whereas the relaxation of paramagnetic Co^2+^ ions is mostly driven by interactions with the paramagnetic center.

## Figures and Tables

**Figure 1 molecules-24-02895-f001:**
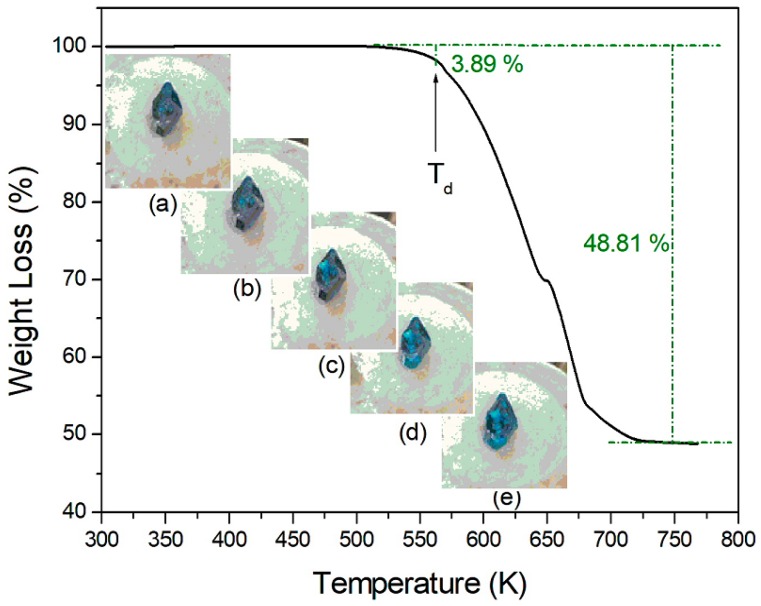
Thermogravimetric analysis (TGA) curve of (CH_3_NH_3_)_2_CoBr_4_ (inset: states of the crystal at temperatures of (**a**) 300 K, (**b**) 400 K, (**c**) 500 K, (**d**) 550 K, and (**e**) 570 K).

**Figure 2 molecules-24-02895-f002:**
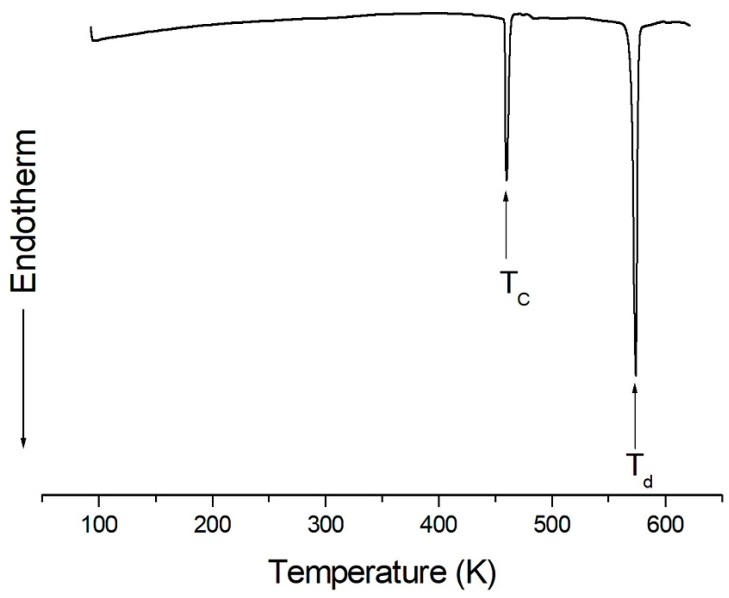
Differential scanning calorimetry (DSC) curve of (CH_3_NH_3_)_2_CoBr_4_.

**Figure 3 molecules-24-02895-f003:**
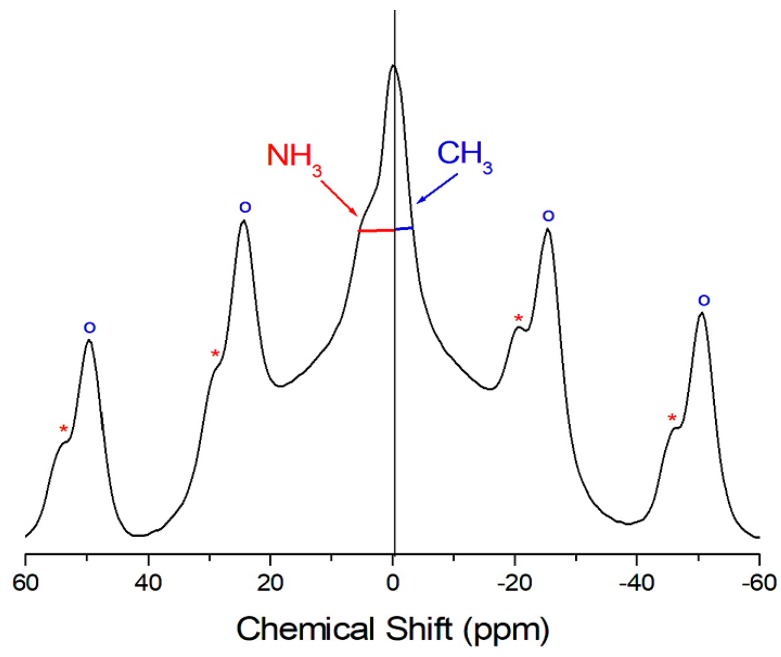
^1^H-NMR spectrum for (CH_3_NH_3_)_2_CoBr_4_ crystal at 410 K. The open circles are the marked sidebands for CH_3_ and the asterisks are the marked sidebands for NH_3_.

**Figure 4 molecules-24-02895-f004:**
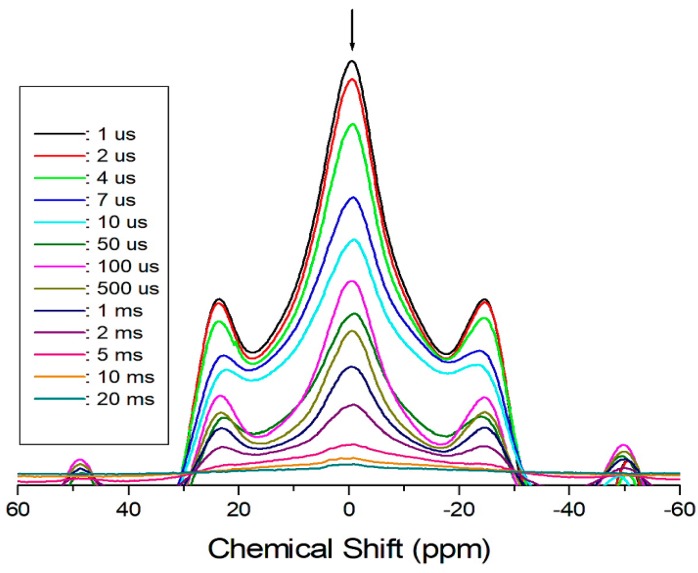
Recovery traces for ^1^H-NMR spectrum in (CH_3_NH_3_)_2_CoBr_4_ as a function of delay time from 1 μs to 20 ms.

**Figure 5 molecules-24-02895-f005:**
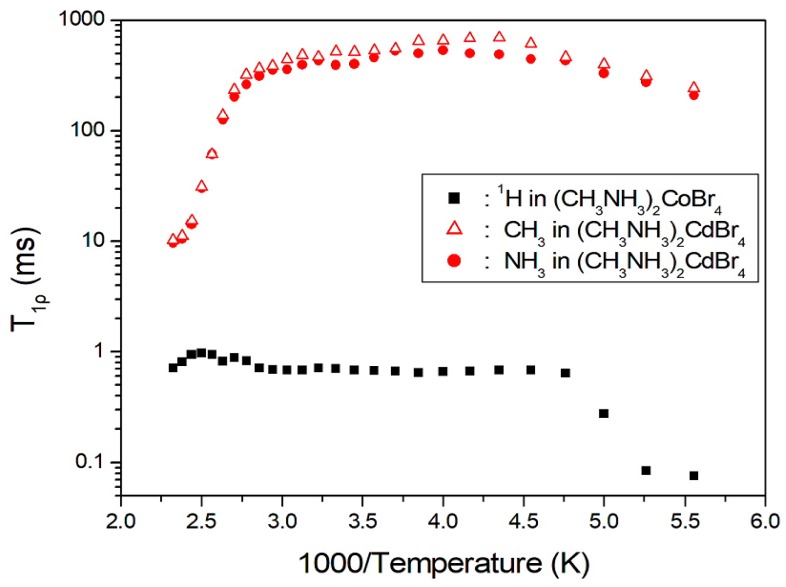
^1^H spin–lattice relaxation times in (CH_3_NH_3_)_2_CoBr_4_ and (CH_3_NH_3_)_2_CdBr_4_ as a function of inverse temperature.

**Figure 6 molecules-24-02895-f006:**
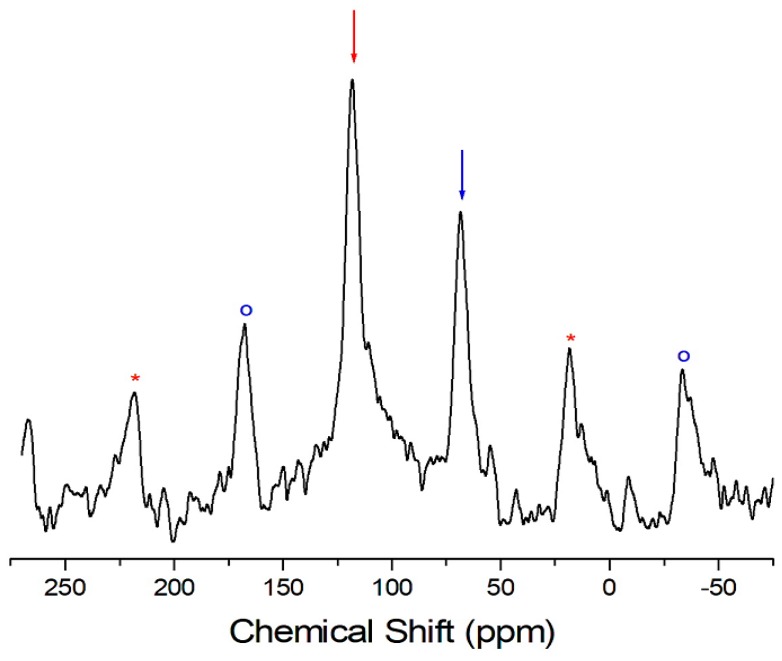
^13^C-NMR spectrum in (CH_3_NH_3_)_2_CoBr_4_ at 300 K. The two arrows denote the signals of the two crystallographically different CH_3_ moieties. The spinning sidebands are marked with open circles and asterisks.

**Figure 7 molecules-24-02895-f007:**
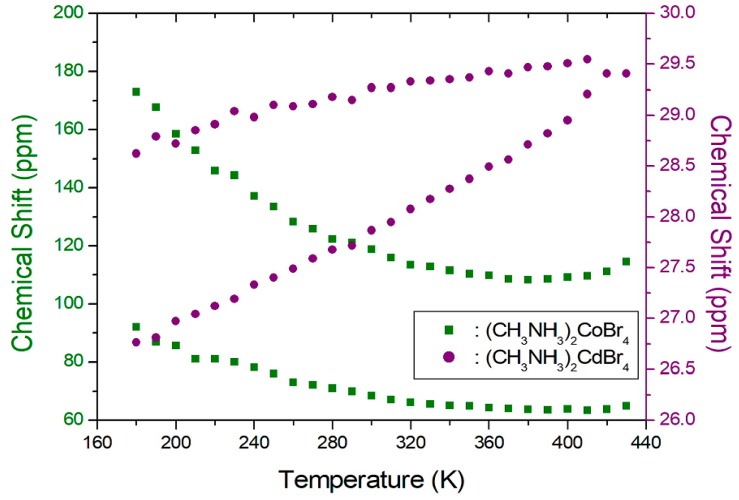
^13^C-NMR chemical shifts of (CH_3_NH_3_)_2_CoBr_4_ and (CH_3_NH_3_)_2_CdBr_4_ as a function of temperature.

**Figure 8 molecules-24-02895-f008:**
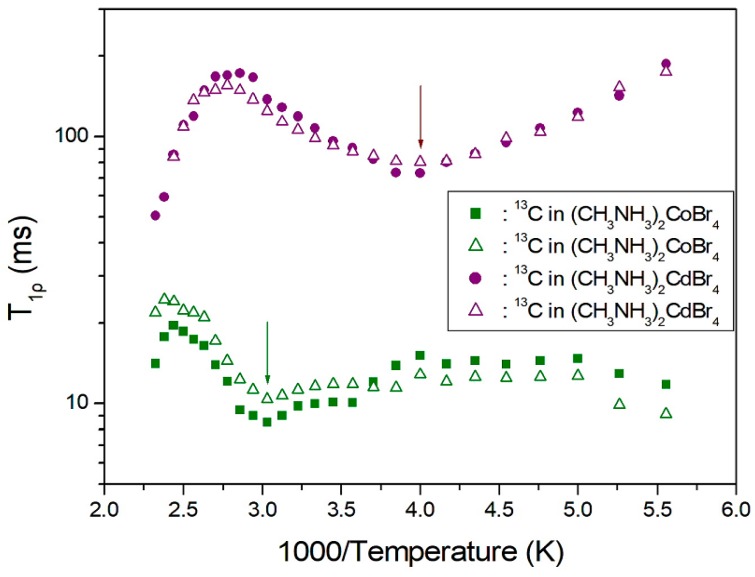
^13^C spin–lattice relaxation times of (CH_3_NH_3_)_2_CoBr_4_ and (CH_3_NH_3_)_2_CdBr_4_ as a function of inverse temperature.

**Figure 9 molecules-24-02895-f009:**
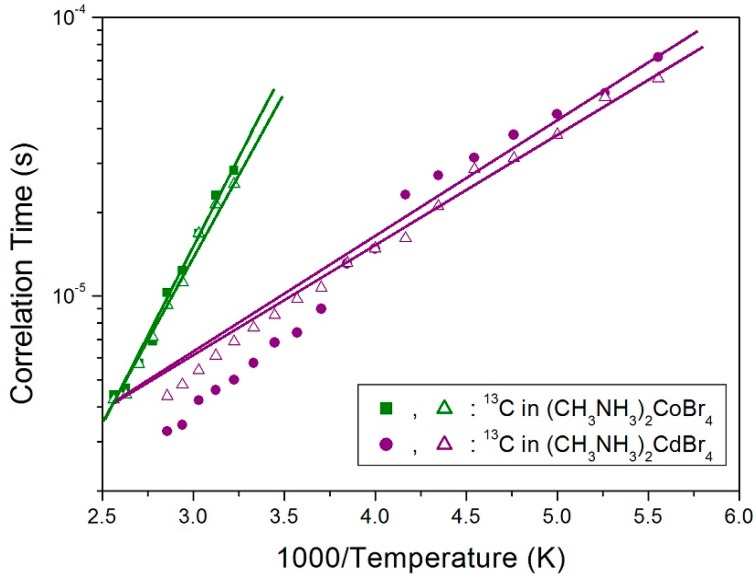
Arrhenius plots of the natural logarithm of the correlation times for each of the carbons of a-CH_3_ and b-CH_3_ in (CH_3_NH_3_)_2_CoBr_4_ and (CH_3_NH_3_)_2_CdBr_4_ as a function of inverse temperature.

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
