# Peer review of "Study on Paramagnetic Interactions of (CH_3_NH_3_)_2_CoBr_4_ Hybrid Perovskites Based on Nuclear Magnetic Resonance (NMR) Relaxation Time"

_molecules, 2019, doi:10.3390/molecules24162895_

Round 1
Reviewer 1 Report
no comments
Author Response
Response: Thank you very much.

Reviewer 2 Report
In this resubmission, the authors corrected the mistakes in the calculations and addressed the questions from previous reviewers properly. However, the design of the study doesn’t make too much sense. The title is “Study of Cationic Molecular Dynamics of…based on NMR Relaxation time”, but it looks like the NMR relaxation measurements were dominated by paramagnetic interaction and didn’t relate to molecular dynamics at all. The significant decrease of the T1ρ and the change of the chemical shifts introduced by paramagnetic centers are predictable. The comparison of the relaxation data of (CH3NH3)2CoBr4 with (CH3NH3)2CdBr4 is not quite meaningful either because of the presence of paramagnetic centers. No information on molecular dynamics of (CH3NH3)2CoBr4 can be extracted from the NMR data. Overall, I don’t think this manuscript is qualified for this journal since the experimental results didn’t support the purpose of the study very well.
Author Response
Response 1: Title “Cationic Molecular Dynamics of (CH3NH3)2CoBr4 Hybrid perovskites based on NMR Relaxation Time” is revised to “Study on Paramagnetic Interactions of (CH3NH3)2CoBr4 Hybrid perovskites based on NMR Relaxation Time”
Response 2: The reviewer considered the key point of the manuscript as a paramagnetic interaction and suggested a good opinion. And it is a good idea to point out that the comparison between (CH3NH3)2CoBr4 and (CH3NH3)2CdBr4 is meaningless. To emphasize the paramagnetic effect, I think that some comparisons are okay. I would like your consideration.
Response 3: “molecular dynamics” in manuscript is revised to “paramagnetic interactions”.
Response 4: The following sentences are deleted because they are not meaningful: “Accordingly, relativity does not play a significant role in the 1H NMR chemical shift, whereas the local field is quite important for the 13C NMR chemical shift.” in Abstract is deleted. “Relativity does not play a significant role in the 1H NMR chemical shift. By contrast, this appears to be important for 13C NMR chemical shifts.” in Conclusion is deleted.

Round 2
Reviewer 2 Report
The authors changed the "molecular dynamic" to "paramagnetic interaction" and modified the conclusion. This change makes sense because the experimental data match the conclusion of the study after the revision. I recommend that the paper should be accepted for publication in its present form.
This manuscript is a resubmission of an earlier submission. The following is a list of the peer review reports and author responses from that submission.
Round 1
Reviewer 1 Report
The authors of this study investigated hybrid perovskites based on NMR relaxation time. In particular, two rarely explored perovskites such as (CH3NH3)2CoBr4 and (CH3NH3)2CdBr4 were tested and compared. In brief, the present article is scientifically sound and can be considered for publication after a revision process.
1) It is highly recommended to revise a manuscript with a native speaker.
2) Authors should provide the XRD measurements of both perovskites prior to TGA & DSC and after TGA & DSC measurements to confirm the transitions proposed in Eq. 1 & 2.
3) It is highly suggested to perform the time-resolved PL study and UV-Vis of perovskites.
Reviewer 2 Report
In this manuscript, Lim and co-workers investigated the thermodynamic property and molecular dynamics of (CH3NH3)2CoBr4 using DSC, TGA, and MAS-ssNMR. The results were compared with the previous study of (CH3NH3)2CdBr4 from the same group. The authors concluded that the dynamics of the (CH3NH3)2+ cations are very different in these two cases because of the paramagnetic nature of the Co2+ ion. The experimental results are very interesting. However, this manuscript is lack of solid conclusion. Some data analysis is wrong or not thorough enough.
Major issues
1. The analysis of τc of (CH3NH3)2CoBr4 is wrong. Equation 4 in the manuscript is based on the relaxation caused by dipolar couplings, so the calculation for τc of (CH3NH3)2CdBr4 is correct. However, when the paramagnetic centers exist (i.e. Co2+), the interaction between unpaired elections and target nuclei dominates the relaxation. In this case, the Solomon equation for the longitudinal relaxation in rotating frame is shown in the attachment, where ge is the gyromagnetic ratio of the election, S is the total spin quantum number for the paramagnetic center, ωe is the Larmor frequency for electron.
2. When the paramagnetic centers exist, 1/τc = 1/τr+ 1/τM + 1/τe, where τr, τM, and τe are rotational correlation time, exchange correlation time, and electronic relaxation correlation time, respectively. τr is the one can represent molecular motion/dynamics. For (CH3NH3)2CdBr4, there is no chemical exchange or paramagnetic terms, so τc can directly reflect the molecular motion. However, for (CH3NH3)2CoBr4, τe dominates the total correlation time. So, I don’t think the T1ρ or τc directly relate to the molecular motion when the paramagnetic centers are present.
3. There are a few interesting data without a clear interpretation. For example, why do the 13C chemical shifts of the two resonance peaks of (CH3NH3)2CdBr4 converge as temperature increase but the (CH3NH3)2CoBr4 is not? Is it because of the pseudocontact shift introduced by paramagnetic centers? In figure 4, why the increase of T1ρ at low temperature only happen for (CH3NH3)2CoBr4 but not for (CH3NH3)2CdBr4?
4. The TGA plot of (CH3NH3)2CoBr4 has been published in another paper as well (Chem. Commun., 2019,55, 6779-6782). However, it looks different from the TGA plot in this manuscript. Please discuss the difference.
Minor issues
1. The authors need to express the significance of the study in the introduction. It is not clear why the paramagnetic center and molecular dynamic of the perovskite is important for the application on solar cells.
2. The T1ρ is spin-lattice relaxation time in rotating frame. The author used spin-lattice relaxation time in this manuscript sometimes. Please correct the y-axes of figure 4 and figure 7 as well.
3. The resolution of figure 1a-e is a little low. The phase transition cannot be distinguished very well.
4. Figure 3 is heavily overlapped and hard to observe the change of the resonance peaks.

Reviewer 3 Report
The manuscript describes an investigation of an organic-inorganic perovskite material by differential scanning calorimetry, thermogravimetry and NMR spectroscopy. Temperature dependence of proton and carbon chemical shifts and T1ρ relaxation times were determined and interpreted in terms of molecular dynamics of the system.
This work is an extension of a previous work of Ae Ran Lim, where the same experimental techniques were used to characterize an analogous system containing diamagnetic cadmium cation as the central atom. The material investigated in the current manuscript contains paramagnetic cobalt cations. The results obtained previously are compared with those obtained for the cobalt material.
The differences in NMR properties of the two materials are mostly consequence of the paramagnetic nature of cobalt cation, i.e. carbon chemical shifts have “unusual” values and are significantly temperature dependent, and relaxation times are significantly shorter. Although the manuscript brings new data about the material, the discussion is too brief and probably not accurate. The NMR relaxation in the diamagnetic material is most probably driven by dipolar interactions, whereas the relaxation in the paramagnetic material will be mostly driven by interactions with the paramagnetic center. Furthermore, cobalt has a spin 7/2 nucleus, i.e. quadrupolar interactions may contribute to relaxation as well. However, this is not stressed in the manuscript and the analysis of NMR relaxation uses the same approach for both systems.
Minor issues:
Figure 2 and 5: NMR spectra are usually presented with opposite chemical shift scale, i.e. going from higher chemical shifts (left) to lower chemical shifts (right).
Figure 4: Both the red circles and red triangles have the same legend (1H in (CH3NH3)2CdBr4). One of them is probably for the CH3 signal and the other for the NH3 signal.
Figure 6: The y axis on the left-hand side should be color coded to correspond with the color coding of the legend.
Figure 8: No data points are shown for the cobalt material at temperatures below ca. 310 K, although relaxation data were measured down to 180 K. Why?
Reviewer 4 Report
This manuscript, entitled « Study of Cationic Molecular Dynamics of (CH3NH3)2CoBr4 Hybrid Perovskites based on NMR Relaxation Time » describes the thermal properties and the 1H and 13C solid-state NMR of organic–inorganic perovskite (CH3NH3)2CoBr4.
This paper is disappointed. It is poorly drafted and then strongly needed to be revised to enhance the results as shown by the numerous remarks below (in order of decreasing importance).
A recent paper, Chem. Commun., 2019, 55, 6779, reports the same thermogravimetric analysis (TGA) of the same compound (CH3NH3)2CoBr4, but is not cited nor discussed. I am not an expert on TGA, but why are your results completely different from Babu at al?
How is made the temperature calibration for low and HT NMR? The sample temperature is always different from the indicating temperature. It is not possible to directly measure the temperature of the sample. For example, the air friction during sample spinning produces a significant temperature increase, which here might be as high as 50 or even more degrees above room temperature.
Figure 2. I have never seen such a bad figure. Why the authors did not apply the base line corrections? Where are the other spinning sidebands?
How did you determine the chemical shift values ? The determination of the chemical shift values at the peak maximum is not correct in case when the signals overlap. You should fit all spectra (the both 1H and 13C), for example, with DMfit program.
What is the measurement accuracy of the chemical shifts? Two digits to the right of the decimal point?!
NMR results should be compared and discussed with TGA and DSC results.
Did you make the assignment of 13C resonances? Explain, please.
“The 1H NMR chemical shifts for the two 1H in the (CH3NH3)+ cations are temperature-independent.” Usually, the paramagnetic shifts are highly sensitive to temperature. Can the authors discuss this point in more detail?
“The 13C NMR chemical shifts vary significantly with temperature.” These results (effects) are not discussed.
The activation energy values should be compared and discussed with the literature data.
Did you try to obtain the 14N NMR spectra? Why?
“The samples were spun at MAS of 10 kHz using N2 gas to minimize any spinning sideband overlap.” It is not clear. The samples were spun at a sufficiently speed to avoid any spinning sidebands overlapping.
“…obtained at the Larmor frequency of ω0/2π=400.13 MHz….” and “…was studied at the Larmor frequency of ω0/2π=100.61 MHz by 13C MAS NMR.” The repetition of the details of the experimental part.
Line 63.” …Bruker 400 MHz NMR spectrometer…” Which? DSX, AVANCE1,2.3, or NEO?
Line 162. “as a function of Temperature.” Should be “as a function of temperature.”